# Exploring the Microbial Ecology of Water in Sub-Saharan Africa and the Potential of Bacteriophages in Water Quality Monitoring and Treatment to Improve Its Safety

**DOI:** 10.3390/v16121897

**Published:** 2024-12-09

**Authors:** Boniface Oure Obong’o, Fredrick Onyango Ogutu, Shauna Kathleen Hurley, Gertrude Maisiba Okiko, Jennifer Mahony

**Affiliations:** 1Food Technology Division, Kenya Industrial Research and Development Institute (KIRDI), Nairobi P.O. Box 30650-00100, Kenya; phasoure@gmail.com (B.O.O.); gettieokiko@gmail.com (G.M.O.); 2APC Microbiome Ireland, School of Microbiology, University College Cork, College Road, T12 K8AF Cork, Ireland; 123101546@umail.ucc.ie

**Keywords:** bacteriophages, water treatment, antimicrobial resistance, water quality, food safety, sub-Saharan Africa, food preservation

## Abstract

Access to safe water and food is a critical issue in sub-Saharan Africa, where microbial contamination poses significant health risks. Conventional water treatment and food preservation methods have limitations in addressing water safety, particularly for antibiotic-resistant bacteria and other pathogenic microorganisms. This review explores the potential application of bacteriophages as an innovative solution for water treatment and food safety in the region. Bacteriophages specifically infect bacteria and offer a targeted approach to reducing bacterial load, including multidrug-resistant strains, without the drawbacks of chemical disinfectants. This review also highlights the advantages of phage bioremediation, including its specificity, adaptability, and minimal environmental impact. It also discusses various case studies demonstrating its efficacy in different water systems. Additionally, we underscore the need for further research and the development of region-specific phage applications to improve water quality and public health outcomes in sub-Saharan Africa. By integrating bacteriophage strategies into water treatment and food production, the region can address critical microbial threats, mitigate the spread of antimicrobial resistance, and advance global efforts toward ensuring safe water for all.

## 1. Introduction

Water and food are essential to human life, and access to safe and sufficient water and food is recognised as a fundamental human right. The United Nations Sustainable Development Goal (SDG) targets 2 and 6.1 emphasise universal and equitable access to safe and affordable food and drinking water, respectively. Despite these goals, approximately 2.2 billion people live globally without access to safe water, predominantly in low- and middle-income countries, mainly in sub-Saharan Africa (SSA) and Asia [1]. About 319 million people in SSA lack access to reliable drinking water sources, and 695 million lack improved sanitation facilities [2]. The contamination of water supplies with microbial pathogens is a significant public health challenge, contributing substantially to the disease burden in these regions [3]. Unsafe drinking water is a key cause of diarrhoea, responsible for approximately 10% of global mortality among children under the age of five [4]. Despite efforts to improve water access, 42% of the population of sub-Saharan Africa still lack access to basic water supplies, defined as improved water sources accessible within a 30 min round trip [5].

Even with improved water sources, water quality remains a concern as many sources remain contaminated. Drinking water in Africa may be derived from various sources, including unprotected rivers, streams, hand-dug wells, and springs, which are susceptible to multiple sources of contamination ranging from floods to birds, animals, and human activities [6]. Although some populations rely on improved and protected sources such as on-plot piped supplies, shared piped water from standpipes and kiosks, protected springs, wells, boreholes, and harvested rainwater, the risk of microbial contamination remains high [7]. The most significant microbial risks are associated with ingesting water contaminated with faeces from humans or animals, leading to infections caused by pathogenic bacteria, viruses, protozoa, and helminths [8,9]. The contamination of natural water resources due to anthropogenic activities, including population growth, agriculture, and industrialisation, exacerbates water quality issues [10]. Discharging physical, chemical, and biological contaminants from non-point and point sources further impairs water quality and stresses aquatic ecosystems [11].

Furthermore, the downstream use of this water in crop irrigation, as drinking water for animals, and in food production increases the risk of consumers ingesting pathogenic microorganisms in the associated foods. Therefore, the implications of poor water quality on the entire food chain are vast and require urgent attention. Figure 1 shows the interaction between water sources, applications, and sources of contamination.

Antimicrobial resistance (AMR) is a growing global health concern. In 2019, bacterial AMR was linked to an estimated 4.95 million deaths worldwide, with *Escherichia coli* (*E. coli*), *Staphylococcus aureus*, *Streptococcus pneumoniae*, *Acinetobacter baumannii*, *Klebsiella pneumoniae*, and *Pseudomonas aeruginosa* among the leading six pathogens responsible for associated mortalities. This number is projected to increase to 10 million by 2025 [12]. Antimicrobial resistance may spread between animals, humans, and the environment due to their interconnectivity, thereby underpinning the rise of these so-called “superbugs” [13]. A One Health approach that incorporates humans, animals, and the environment needs to be applied to solve the AMR challenge. To achieve this, antibiotic use must be examined in humans and animals while evaluating the environmental impact. Although antibiotic prescription is usually reserved for clinical infections in humans and companion animals, the same applies to food-producing animals. Among chicken flocks and pens of pigs, antibiotics are often administered to whole groups and have (historically) been used as growth promoters [14]. This has facilitated the development and spread of AMR to humans through the environment or animal food products. This continues to significantly impact global food security, food safety, and the economic global food market. In 2022, the European Union banned the use of antibiotics in animals as growth promoters; however, this is not universally applied and may become an international standard in the future, which would reduce antimicrobial residue transmission through the food chain.

## 2. Ecology of Sub-Saharan African Water Systems

South Saharan Africa’s water systems, including freshwater lakes, rivers, wetlands, and estuaries, are biodiverse and are crucial in maintaining ecological balance and supporting the region’s economic livelihood. The main water sources in the region are (a) rivers, such as the Nile and the Zambezi; (b) lakes, including Lake Victoria, Tanganyika, and the Rift Valley lakes, as well as the Congo basin and the Okavango Delta; and (c) numerous groundwater aquifers. However, the ecology of these water sources has been significantly altered, primarily due to human activity. The destruction of natural water resources has been accelerated by human activities such as population growth, agriculture, industry, and other socio-economic development [10]. The quality of surface water is further compromised by the increased discharge of physical, chemical, and biological contaminants, which also threaten aquatic life. These contaminants originate from non-point sources (including surface run-off, airborne pollutants, and sewage outflows), point sources (including industries and direct effluent disposal), and hydro-morphological sources (natural processes and human activities, including water abstraction) [11]. 

Seasonality strongly influences the availability and quality of water in SSA; dry and rainy seasons dictate water access and quality [2]. During the rainy seasons, abundant water is supplied from replenished surface water sources and aquifers. Conversely, during the dry season, diminished water availability becomes widespread following the shrinking of surface water sources and dropping groundwater levels. The reduction in water volume can result in an increased concentration of microbial contaminants in the reduced water volume, increasing the risk of waterborne diseases [15,16]. Conversely, during rainy seasons, surface run-off and floods can lead to a spike in microbial load and dissemination, especially in areas where open defecation is practised [17].

Many studies across the region have revealed the grim reality that microbial contamination is a pervasive and severe issue, impacting a range of water sources. Among these, rivers and lakes, the lifeblood of many communities, are the most affected. A striking example is Lake Victoria, the world’s third-largest freshwater lake, which supports over 42 million people. This vital resource suffers from alarmingly high levels of bacterial contamination due to industrial and wastewater discharge, including pharmaceutical residues, heavy metals, agrochemicals, and personal care products [18]. Metagenomic analysis of the microbial assemblage at Winam Gulf (a northeastern extension of Lake Victoria) from the flood plains of the inlet rivers, wastewater treatment plant discharge points, industrial effluent areas, fish landing beaches, storm water entrance points, and various lake and river locations revealed the following relative abundance of the top five bacterial taxa at the genus level: *Acinetobacter* (18.9%), *Arcobacter* (14.1%), *Flavobacteriales* (7.3%), *Dechloromonas* (6.6%), and *Pseudomonas* (5.5%) [19]. Four of these five genera and their associated species are known human or animal pathogens, being the etiologic agents associated with human lung infections (*Acinetobacter* [20], *Pseudomonas* [21]), urinary tract infections (*Acinetobacter*) [22], diarrhoea (*Arcobacter*) [23], and skin lesions and gill necrosis in freshwater fish (*Flavobacterium*) [24]. The findings underscore the potential risks associated with the consumption of this water and its application in the food production chain. *Vibrio cholera*, a bacterium synonymous with its namesake illness, cholera, has also been reportedly isolated from water samples collected from various sources in the Lake Victoria region of Kenya. This is a stark reminder of the immediate threat to public health and the need for immediate action [25,26]. In a study on the enteric bacteria isolates from water and fish, the prevalence and antimicrobial susceptibility patterns showed *Salmonella typhimurium* as the most prevalent pathogenic bacteria (49.6%), followed by *Escherichia coli* (46.6%) and *Vibrio cholerae* (2.8%) using culture-based methods, with all isolated bacteria exhibiting antibiotic resistance [27]. 

Groundwater, which serves as the primary drinking water source for a large portion of the rural population in SSA, is also not immune to microbial contamination. Shallow wells [28] and boreholes [29,30] are frequently contaminated with pathogenic bacteria and coliforms at levels exceeding World Health Organization (WHO) guidelines. In informal settlements such as slums and refugee camps with no formal sanitation systems, drinking water infrastructure often relies on pit latrines and shallow wells for domestic water use and waste disposal [31]. Often, the placement of the two does not meet the minimum required distance of at least 30 m, or the latrine is placed downhill or at a lower elevation than the well, leading to contamination through groundwater flow [32]. Also, improper pit latrine construction or maintenance can lead to groundwater contamination. When pit latrines are not adequately sealed or lined, faecal matter and pathogens can seep into the surrounding soil, potentially contaminating groundwater sources. These unplanned settlements and associated infrastructure pose significant health risks to communities relying on groundwater for drinking water [33,34]. Seasonal variations exacerbate the situation. During rainy seasons, surface run-off and floods can lead to a spike in microbial load, especially in areas where open defecation is practised, while the dry seasons may see a concentration of microbial contaminants in diminished water flows, increasing the exposure risk to local populations [15,16].

## 3. Water Quality in Sub-Saharan Africa

Unsafe drinking water is a significant cause of diarrhoea, responsible for approximately 10% of global mortality among children under the age of five. Waterborne microbial pathogens are responsible for a significant portion of the public health burden affecting low-income countries [4]. Reports indicate that diarrhoea accounts for 1.6–2.5 million deaths annually and remains one of the main causes of morbidity in developing countries, with each child experiencing an average of three episodes of diarrhoea per year [35]. Infectious diarrhoea is a common disease among under-fives. The responsible agents are viruses, bacteria, protozoa, and helminths. Bacteria are responsible for about 45% of diarrhoeal illnesses. *E. coli*, *Campylobacter jejuni*, *Shigella*, *Salmonella*, and *Vibrio El Tor* are among the top-ranking bacterial causative agents of diarrhoea. Water has been linked to antimicrobial resistance (AMR) development in antimicrobial-resistant strains of waterborne pathogens, including *Pseudomonas aeruginosa*, *Mycobacterium* spp., and *Legionella* spp., and methicillin-resistant *Staphylococcus aureus* and carbapenem-resistant *Enterobacteriaceae*, which are not typically associated with water, were also isolated. Further, biofilms were identified as a hot spot for the transfer of genes responsible for survival functions [36]

The Guidelines for Drinking Water Quality (GDWQ) assess the health risks of various microbiological, chemical, physical, and radiological constituents in drinking water [8]. The resulting guidelines describe the reasonable minimum requirements of safe practice to protect consumers’ health while also providing numerical “Guidelines Values” for certain water constituents or indicators of water quality [37]. The primary purpose of the GDWQ is geared towards protecting public health by ensuring access to safe drinking water. These guidelines must be contextualised within local or national environmental, social, economic, and cultural conditions to be effective. They form part of a broader health protection strategy that includes sanitation and measures to manage food contamination [8]. In many countries, drinking water safety management involves two key activities: operational monitoring by licensed water suppliers and compliance monitoring by an independent public health agency. Operational monitoring ensures the effectiveness of water treatment and distribution processes and guides necessary corrective actions. Compliance monitoring, on the other hand, involves the oversight of regulated water supplies and assessments of informal and community-managed sources [37].

An evaluation of the microbiological and physico-chemical quality parameters of water in Guinea-Bissau found that most water sources used by the population were grossly polluted with faecal matter (80%) and hence unsuitable for consumption. The highest levels of faecal pollution were observed during the wet season, with hand-dug wells having the highest contamination levels [2]. A similar analysis of the physico-chemical water quality in the Wondo Genet campus in Ethiopia found that the water met the WHO specifications for drinking water, highlighting the regional variation in water quality [6]. To establish drinking water testing programs for faecal contamination in 72 institutions (water suppliers and public health agencies) across ten countries in sub-Saharan Africa, 85% of institutions had conducted microbial water testing in the previous year to varying extents. The study concluded that small-scale water providers, including rural public health offices, require greater attention and added resources to achieve regulatory water quality monitoring and compliance in sub-Saharan Africa [4].

A comprehensive study investigated 21 water quality parameters across 18 selected African countries, whose findings established the variability in water quality standards across the continent. This study found significant differences in 20 of the 21 parameters, with water quality standards in these African nations generally weaker than those set by global benchmarks such as the WHO, EU, and China. The assessed parameters included biological factors (e.g., *E. coli* and total coliforms), chemical components (e.g., nitrates, nitrites, and fluorine), physical attributes (e.g., total solids, turbidity, and conductivity), toxic heavy metals (e.g., chromium, lead, and arsenic), inorganic toxic chemicals (e.g., nitrates and nitrites), and organic constituents (e.g., benzene) [38]. The findings emphasise a critical need to develop regional water quality standards and practices in Africa, as current national regulations are insufficient to ensure safe drinking water, particularly in comparison to international standards. This reinforces the need for targeted efforts to improve water quality and monitoring, especially in regions where current standards are inadequate to protect human health. The water quality in sub-Saharan Africa has been associated with several pathogenic microbes, as shown in Table 1 below.

Access to safe and quality water is critical for improving health and livelihoods in low-income communities across sub-Saharan Africa (SSA). According to the WHO, a risk–benefit approach to establishing drinking water standards is advantageous. This approach focuses on balancing the risks and benefits to ensure that resources are used efficiently and that standards protect public health and are feasible to implement [37]. In SSA, water quality has not been extensively studied due to the lack of automated monitoring systems that provide reliable and frequent data. Nonetheless, it is estimated that 40% of the 320 million people in Africa who lack access to safe drinking water reside in this region [52]. The scarcity of clean water sources and the prevalence of waterborne pathogens underscores the need for innovative solutions, such as applying bacteriophages, to enhance water quality and safeguard public health [53].

### 3.1. Water Treatment Practices

Water quality concerns, mainly water destined for drinking and domestic purposes, have resulted in various water treatment technologies that purify water to varying degrees. The employed treatment methods depend on the quality of the raw water source. A single process or a chain of treatment processes are adopted to produce water of desirable quality [54].

The most common water treatment processes around the world are rapid mixing (hydraulic mixers), flocculation (hydraulic and mechanical), sedimentation/clarification (rectangular horizontal flow sedimentation), filtration (rapid sand), and disinfection (chlorination) [10]. The suspended and colloidal impurities in water are removed in steps by rapidly mixing coagulants or the flocculation and sedimentation processes [55]. Colloidal and other finer particles, apart from dissolved organic and microbial impurities, can also be removed by filtration techniques (Figure 2).

### 3.2. Global Water Treatment Processes

In Canada, raw water flows from rivers to water treatment plants (WTPs) across the country. Then, it is pumped to a WTP through a reservoir and undergoes several steps of treatment to produce high-quality water [56]. The treatment objective is to exclude and reduce enteric viruses, pathogenic bacteria, *Giardia* cysts, and *Cryptosporidium* oocysts to safe levels. The objectives are as follows: 4-log reduction or inactivation of viruses; 3-log reduction or inactivation of *Giardia* and *Cryptosporidium*; and no detectable *E. coli*, faecal coliform, and total coliform [57]. To achieve microbiological destruction, WTPs in Canada have added ozonation and ultraviolet (UV) treatment steps in their water treatment process. Ozonation is performed to kill bacteria, and sodium bisulphite is added to remove excess ozone. In Japan, water treatment processes employ conventional steps supplemented with ozone treatment and biologically activated carbon treatment [58]. The ozone purification process lasts 20 min, and fine bubbles are diffused slowly into the water to increase efficiency and ensure more ozone dissolves in the water. The process then enters the second stage of advanced water treatment by flowing the water into a filtration pond containing different grades of biological activated carbon [58].

The EU Drinking Water Directive 2020/2184 of the European Parliament and Council is the primary law for drinking water quality in the European Union. Each EU member state is responsible for preparing drinking water and reporting the water quality standards. Ground and surface water are the major water sources, and they are treated using various methods depending on the member state. These treatments include conventional treatments as described above, which may be replaced or supplemented with carbon filtration, advanced oxidation processes, rapid sand filtration or aeration [59].

### 3.3. Water Treatment Process in Sub-Saharan Africa

Across SSA, most countries heavily depend on conventional water treatment approaches to supply clean drinking water to their populations [60]. While this reliance underscores the urgent need for innovative solutions in the water treatment sector, it also presents a promising opportunity for improvement. For instance, in East and South Africa, the widely used method for drinking water treatment involves coagulation and flocculation using chemical coagulants like aluminium sulphate (alum) to remove dirt, organic materials, and microorganisms. This is followed by sedimentation, where the water is held in large tanks, allowing flocs to settle at the bottom, separating the solid particles from the water. The water is then filtered through sand filters (slow and rapid) made of sand, gravel, and charcoal to remove remaining particles, pathogens, and some chemical contaminants. Subsequently, chlorine may be added to the water to eliminate microbial contaminants. The treated water is stored in tanks for further pipeline distribution [60].

The major drawback of chlorine in water treatment is the formation of disinfection by-products (DBPs) [61], which have been linked to cancer and other health issues following long-term exposure to high levels of these by-products [62]. Chlorine has also been reported to be less effective against certain protozoa, such as *Cryptosporidium* and *Giardia*, which have cyst forms resistant to chlorination [63]. Also, chlorine and its by-products can significantly affect aquatic ecosystems if they enter natural water bodies [64].

Therefore, for the effective microbial decontamination of water during conventional-process treatment, additional treatment methods, like membrane filtration, UV disinfection, or ozonation, should be considered to handle chlorine-resistant pathogens. Phage deployment is also a promising inclusion into the process, either as cocktails or for targeted organisms. One notable example is the use of cyanophages to control toxic cyanobacterial blooms, such as those caused by *Microcystis aeruginosa*, in surface drinking water sources. These cyanophages specifically target cyanobacteria, offering a precise method to reduce harmful cyanotoxins without necessitating substantial changes to the existing water treatment infrastructure [65].

Pollution is another challenge facing the conventional water treatment process that is widely adopted in SSA. Contamination from industrial and agricultural activities can complicate treatment processes due to chemical pollutants like Cr, Zn, As, Ag, Cd, and Pb [66]. Also, wastewater treatment plants are a significant source of antibiotic residues and antibiotic-resistant bacteria in the environment [67], highlighting the need for improved water treatment methods and more stringent environmental quality standards and monitoring.

## 4. Strategies to Improve Water Quality in Sub-Saharan Africa

Improving water quality in SSA is a multifaceted task that necessitates community engagement, infrastructure investment, and the innovative application of technology, among other factors [68,69]. The methods most municipalities and water treatment companies employ in SSA for treating drinking water are diverse. Some are modern and effective, while others may be outdated or insufficient [70]. These methods typically involve physical, chemical, and biological processes like coagulation, flocculation, filtration, and disinfection, as well as post-treatment steps such as pH adjustment and fluoridation [60]. However, these methods face several challenges, including ageing or inadequate infrastructure that hinders effective water treatment, financial and technical limitations restricting the implementation of advanced treatment methods, variable quality of raw water sources, and contamination from industrial and agricultural activities.

In the face of these challenges, notable successes have been driven mainly by non-state actors’ efforts. Programs targeting household water treatment have effectively reduced waterborne illnesses [71]. For instance, a community-led total sanitation program has been transformative in Kenya. This initiative, spearheaded by an NGO, empowers communities to eradicate open defecation through collective behaviour change, leading to a considerable reduction in open defecation and a marked improvement in water quality by reducing the faecal contamination of water sources. Similarly, in Malawi, the NGO Water for People has spearheaded extensive borehole drilling and rehabilitation projects, ensuring long-term access to clean water for rural communities.

Collection point water treatment strategies have also been successfully deployed. These strategies employ simple but effective water treatment methods like chlorine dispensers, ceramic filters, and solar disinfection. In Kenya, Uganda, and Malawi, chlorine dispensers and portable ceramic filters have been distributed to low-income communities to treat water at the collection point, improving water safety significantly [70]. Other strategies have been implemented in the region, as shown in Table 2.

To provide adequate and quality water, governments in SSA should adopt strategies and approaches encompassing robust and resilient infrastructure development, curriculum-embedded education, policy enforcement, technology innovations, integrated water resource management, research, and climate change adaptation. Rainwater harvesting, coupled with the development of new and upgraded water treatment facilities, could significantly enhance the availability and safety of water to the affected population [72]. Embedding water hygiene and safety into education curricula and training local personnel in water quality monitoring are crucial [73]. Strict enforcement of environmental regulations, especially regarding industrial discharge, agricultural run-off, and sanitation, coupled with developing and reviewing water quality standards informed by research, can prevent contaminants from entering water sources [74]. Water recycling practices like reusing wastewater from domestic activities for non-potable purposes (greywater) and highly contaminated wastewater from toilets and kitchens (blackwater), industrial water recycling, and integrated watershed management are essential to protect and sustain water resources [75]. Adequate funding allocation for water and sanitation by governments and fostering public–private partnerships can ensure culturally appropriate and sustainable solutions [8]. Most importantly, adopting climate-resilient strategies and developing those that help to adapt to climate change impacts are vital for long-term water security in sub-Saharan Africa [76].

Government policies and initiatives are crucial to achieving water safety and security in sub-Saharan Africa. Many countries in the region have implemented national water policies to address water scarcity, quality, and management challenges as they recognise the importance of water quality in improving public health and economic stability. For instance, South Africa, Kenya, and Uganda have made impressive progress in establishing national water policies. For example, South Africa’s National Water Act is considered one of the most progressive in the region, emphasising sustainable and equitable water use. Similarly, Kenya and Uganda have enacted policies on integrated water resource management and improving access to clean water and sanitation [77]. However, other countries have lagged in formulating and implementing their water policies. In Mali, Niger, and Burkina Faso, socio-political instability has complicated the effective implementation of their national water strategies [78].

International partners such as the World Bank, World Health Organisation (WHO), and the United Nations Environment Program (UNEP) and non-governmental organisations like the United States Agency for International Development (USAID) are playing a key role in tackling the problem of water safety and security in the sub-Saharan region. They have collaborated with various country governments, rolling out programs to ensure sustainable water and sanitation management and the realisation of SDG6. Examples of non-government organisations working on collaborative projects in the region include the Millennium Water to Ensure Sustainable Water and Sanitation Management Alliance (MWA) working in Ethiopia, Kenya, and Uganda; WaterAid in Malawi; the Rural Water Supply Network; One Drop Foundation; the World Bank’s Water Global Practice; UNICEF’s WASH Programs; and the Coca-Cola Africa Foundation’s Replenish Africa Initiative (RAIN and USAID’s Sustainable Water and Sanitation in Africa (SUWASA)), among others [79].

## 5. Bacteriophages in Sub-Saharan Africa Water Systems

Bacteriophages are viruses that specifically infect strains of a given bacterial species. Many have a narrow host range (i.e., they infect a small number of strains of a given bacterial species), while others have a broad host range. They are the most abundant and ubiquitous organisms worldwide and can be found in natural and artificial environments, especially in which their bacterial host thrives [80]. Among the bacteriophages that have been isolated, there are numerous reports of phage screening studies specifically located in SSA, particularly in Nigeria, Sudan, Malawi, Egypt, South Africa, Namibia, Kenya, and Cameroon, from various environmental sources like sewage, water, soil, wildlife carcasses, and hot springs [81]. These phages target a variety of bacteria, including *Vibrio cholerae* [26], *Salmonella typhimurium*, *S. enteritidis* [82], *Pseudomonas aeruginosa* [83], *E. coli* [84], *Ralstonia solanacearum* [84], *Bacillus anthracis* [85], *Staphylococcus aureus* [86], and *Mycobacterium smegmatis* [87]. The isolated phages were morphologically diverse, representing, in addition to the myovirus, siphovirus, and podovirus morphotypes, the families *Ackermanviridae* and *Fuselloviridae* [88]. It should be noted that the taxonomy of phages no longer follows these morphotypes and is based on genetic compositional analysis [89]. However, since the genomes of the isolated phages were not sequenced in this study, the genetic diversity is currently unknown.

Two primary culture-based approaches are typically applied to isolate and characterise phages. The classical approach involves mixing a sample of phage filtrate with a culture of susceptible bacteria and incubation followed by observational/turbidity measurements to monitor cellular lysis, indicating the presence of infectious particles. This may be performed with plate-based methods such as spot testing, plaque assays, or routine test dilution (RTD) to detect, visualise, and enumerate isolated phages. For the characterisation of phages, pure strains of phages are obtained through multiple rounds of phage plaque purification, followed by extracting sufficient DNA for analysis [90]. Host range testing is also critical in determining the usefulness of phage for phage therapy/biosanitation purposes. This involves testing the isolated phage against a panel of multiple host species/strains to determine which can support phage infection and new virion production.

More recently, such studies have often partnered with metagenomics analysis to evaluate the presence and diversity of phages in environmental samples [91,92,93]. Metagenomic phage identification includes DNA extraction, sequencing, bioinformatic analysis, the identification of the phage, annotation and functional analysis, comparative genomics, and phage identification and characterisation. Metagenomic characterisation has the advantage of not requiring the laboratory isolation and cultivation of phages. However, phage isolation and characterisation remain the gold standard in combination with genomic approaches to explore the viral diversity and abundance, mainly where suitable hosts are not available to propagate all phages [94].

Bacteriophages have been detected across SSA water systems [95,96,97]. In South Africa, phages were isolated from water samples collected from taps, boreholes, and dams targeting *Vibrio* species, including *V. harveyi*, *V. parahaemolyticus*, *V. cholerae*, *V. mimicus*, and *V. vulnificus.* A study on water samples from Malawi and the UK to isolate phages that targeted *Salmonella* variants associated with bloodstream infections found that most of the isolated phages could infect the *Salmonella typhimurium* D23580 ∆ᶲ∆ brex mutant.

In Kenya, a study of bacteriophages from the haloalkaline Lake Elmenteita, using indigenous bacteria as hosts, reported the isolation of 17 different phages with myovirus, siphovirus, and podovirus morphotypes [96]. Similarly, myoviruse and siphoviruses infecting *Bacillus* and *Paracoccus* species were isolated from haloalkaline Lake Chala soil sediments in Ethiopia [98]. Phages isolated from the environmental waters of the Lake Victoria region of Kenya were tested against the isolated *Vibrio cholerae* bacteria to determine their lytic activity. Plaque assays revealed plaque concentrations ranging from 16 to 36 pfu/mL, with complete lysis inferred from the plaques observed. The study suggested the use of the myoviruses as potential biocontrol agents for combating pathogenic *V. cholerae* in water reservoirs in the region [99].

Phages offer a promising alternative to antibiotics in maintaining food safety and curbing zoonotic diseases in SSA [81]. Phages can be used as additives in animal feed to prevent or control bacterial infections, reducing livestock contamination risk. In poultry farming, phage therapy has effectively reduced the prevalence of *Salmonella*, *E. coli*, and *Campylobacter* [100]. Phages can potentially be cost-effective compared to antibiotics, and their production is relatively low-cost. However, there are challenges including limited infrastructure and resources for phage isolation, production, and quality control, unclear or non-existent regulatory frameworks for phage therapy, and limited availability of diverse phage libraries [81]. Addressing these challenges will require multistakeholder efforts between researchers, healthcare professionals, policymakers, and regulatory authorities to increase awareness, develop infrastructure, establish clear regulations, promote research, and address costs to make phage therapy commonplace in SSA. Table 3 summarises a list of phages that have been isolated from water sources in SSA.

## 6. Use of Bacteriophages to Identify Pathogenic Bacterial in Sub-Saharan African Water Systems

Bacteriophages may be promising biomarkers and detection tools in water quality management, especially in water treatment in regions like SSA, where water quality is a big problem. Phages’ specificity and host range (narrow) allow for precise detection of pathogenic bacteria in water sources, which is crucial for ensuring food and water safety. Advancements in phage-based detection methods have demonstrated the efficacy of phages in accurately identifying waterborne pathogens with great precision. In contrast, most common approaches used in the detection of pathogens, e.g., agar plate cultures, enzyme-linked immunosorbent assays (ELISAs), and PCR technology, are time-consuming and take from a few hours to a few days, have high costs, are technically complex, may have limited sensitivity or specificity problems, and present challenges in differentiation between viable and non-viable cells [115]. Recently, bacteriophages have increasingly been deployed as bioprobes in microbial detection because of their high specificity and affinity, low cost, robustness, and high stability [115]. For instance, Anany and colleagues [116] developed a phage-based “dipstick” assay that provides rapid and ultrasensitive detection of foodborne pathogens, which can be adapted for waterborne pathogen detection. The approach uses a combination of phage amplification and qPCR to detect bacterial pathogens; the method demonstrated high sensitivity for various pathogens, including *E. coli* and *V. cholerae*. Another innovative method involves phage lytic assays combined with bioluminescence detection. The method monitors the release of intracellular components such as ATP after phage-induced lysis, providing a rapid and sensitive measure of bacterial concentration in water samples [117].

Phage applications have also been deployed as real-time monitoring tools in water treatment facilities. The approach relies on phage-based reporters in which genetically engineered phages are used to carry reporter genes, which may lead to measurable signals upon infection in bacteria [118]. The approach was applied to specifically detect *Salmonella* in water samples, providing a rapid and reliable means of contamination assessment [119].

In SSA, the implementation of phage-based detection approaches is still in its infancy. A notable example is the application of phages in detecting and neutralising *Vibrio cholerae* in cholera-endemic regions of Lake Victoria, Kenya [26]. Also, phages specific to *V. cholerae* have been isolated and employed in pilot studies, demonstrating their potential in early warning systems for cholera outbreaks [120]. Table 4 summarises studies in which phages were applied in water quality monitoring.

## 7. Water–Food Safety Nexus

Many waterborne pathogens tend to be foodborne, too, as water is critical in food production and processing systems. Water quality is an important pre-harvest factor for preventing foodborne contamination during food production. For example, irrigation water quality can affect food safety and has been identified as a possible source of microbiological contaminants in produce linked to disease outbreaks, including microbes like bacteria (*Salmonella* spp., *E.coli*, *Vibrio cholera*) and viruses like hepatitis [136].

According to CDCP USA, some waterborne diseases include *Vibrio parahemolyticus*, associated with seafood; Norovirus, associated with raw or undercooked shellfish and water; and many others, such as *Escherichia coli*-enterotoxigenic (ETEC) and *Escherichia coli* enteroinvasive (EIEC), both associated with uncooked vegetables, salads, water, and cheese. *Vibrio cholera* is associated with water and shellfish, and enterohemorrhagic *Escherichia coli* (*E. coli* O157:H7 and others) is associated with beef, raw milk, water, apple cider, and lettuce [137]. Enteric viruses are common waterborne pathogens from contaminated water bodies. They include adenovirus, rotavirus, noroviruses, and other caliciviruses and enteroviruses like coxsackievirus and polioviruses. Some viruses have been linked to gastroenteritis, while other enteric viruses have also been implicated in more severe infections such as meningitis, encephalitis, hepatitis (hepatitis A and E viruses), cancer, and myocarditis (enteroviruses) [138].

A microbial quality evaluation of water in Kampala, Uganda was performed based on the detection of genes highlighted the presence of enterohemorrhagic *E. coli*, *Shigella* spp., *Salmonella* spp., *V. cholerae*, and enterovirus [139]. Water is used in various stages of food processing and may also be added to foods. Therefore, the industrial water quality requirement is to have potable water that conforms to physico-chemical and microbiological quality parameters; otherwise, using contaminated water is a risk factor for food- and waterborne diseases. Waterborne diseases remain a major public health and environmental concern, and waterborne diseases continue to plague developing countries, with Africa and Asia being the worst-hit. This is exacerbated by the unavailability of piped water and the high dependence of rural dwellers on surface water sources that are often contaminated with faecal matter. Water unavailability and poor hygienic practices amongst rural dwellers are also of critical concern, as they play key roles in spreading water-washed diseases [140].

### Role of Bacteriophages in Food Safety

Bacteriophages are also increasingly recognized as effective biocontrol agents in the food industry. They target specific bacterial pathogens, reducing the risk of foodborne illnesses [141]. For instance, they have been shown to effectively lower levels of *Salmonella* and *Campylobacter* in poultry, where a reduction of just two logs of *Campylobacter* loads could significantly decrease the incidence of related illnesses [142]. Listex P100, a phage preparation against *Listeria monocytogenes*, has been proven more effective than standard antibacterials like nisin and sodium lactate in eliminating *Listeria monocytogenes* on ready-to-eat sliced ham. Phages specific to Shiga toxin-producing *E. coli* (STEC) were used to reduce bacterial contamination in fresh cucumbers [143] significantly. Moreover, a phage cocktail reduced *E. coli* O157:H7 contamination in lettuce and beef, though it did not protect against recontamination. Phages EP75 and EP335 efficiently reduced viable cell counts of *E. coli* O157 in beef and vegetables. A polyvalent phage was also capable of controlling *Salmonella* and *E. coli* O157:H7 in different food matrices [144].

In addition, phages can be applied directly to food surfaces or processing equipment to control biofilms formed by spoilage and pathogenic bacteria. Bacteriophages JG004 and P1 were tested for effectiveness in reducing *Pseudomonas aeruginosa* on surfaces, resulting in insignificant reductions in bacterial counts on contaminated surfaces, including biofilms [145]. Phage fHe-Yen9-01 was also evaluated for its efficacy against *Yersinia enterocolitica* on kitchen utensils. It reduced bacterial counts significantly when applied to contaminated surfaces [143]. Bacteriophages specifically target and lyse bacteria, providing a selective method of decontamination that minimizes the impact on beneficial microbiota compared to broad-spectrum disinfectants.

Phages are utilized in post-harvest treatments to prevent bacterial growth on fruits and vegetables, enhancing their safety and shelf life also. For instance, a four-phage cocktail reduced *Salmonella* counts by ~3.5 logs on melon fruit slices stored at 5 and 10 °C, and ~2.5 logs at 20 °C, though no reduction was noted on apple slices. Some phages, however, have been used to eliminate *Listeria monocytogenes* from cut melon, apples, and soft cheeses with surface-ripened rinds [142]. A lytic bacteriophage cocktail inactivated *Salmonella enterica* on post-harvest cantaloupe and lettuce [145]. *Salmonella* phages have been used to significantly reduce contamination levels on cantaloupes, and phages specific to Shiga toxin-producing *E. coli* (STEC) were also used to reduce bacterial contamination in fresh cucumbers [146].

## 8. Bacteriophage Application in Water Treatment

The potential points of application of bacteriophages in water cycle is as shown in Figure 3. Below. Biological intervention strategies such as phage-based treatments are considered the future of drinking water treatment, especially in developing countries [10]. Bacteriophages are finding wide applications in phage therapy, vaccine carriers, gene delivery, food preservation and safety, biofilm control, and surface disinfection [147,148]. With the increasing identification of bacteriophages in water, including sewage water, their significance and application potential are worth exploring [147,149]. While studies indicate phages could “solve” the crisis of antibiotic-resistant bacterial prevalence [147], it must not be negated that bacteriophages may contribute to antimicrobial resistance through the transmission of antimicrobial resistance genes (ARGs) by horizontal gene transfer [150]. Therefore, phages selected for therapeutic or bioremediation purposes should be carefully evaluated to minimize such risks and to exclude the application of phages whose genomes harbour antibiotic resistance genes or virulence factors, among other traits. Additionally, it should be considered that phage-resistance may develop among sub-populations if repeated applications of the same phages are used. To alleviate this risk, it would be prudent to use large phage cocktails or rotations of different cocktails. Narrow-spectrum phages are preferred for controlling pathogens in live animals to prevent harm to beneficial bacteria [144].

Bacteriophages that infect human gut bacteria were used as indicators of faecal/viral water pollution and as source tracking markers and can be applied in water quality legislation. Other potential applications of enteric phages include controlling bacterial pathogens in sewage or undesirable bacteria that impede the efficacy of wastewater treatments through biofilm formation on membranes [151].

Recent studies highlight the promising role of bacteriophages in addressing bacterial challenges in wastewater treatment, particularly in water produced from fracking. The application of phages has demonstrated effectiveness in completely inactivating resilient bacteria such as *Pseudomonas aeruginosa* and *Bacillus megaterium* [147]. This underscores the potential of phage therapy as a cost-effective, rapid, and environmentally friendly alternative to traditional methods, which often struggle against bacterial resistance. Phage application as a bioremediation tool offers innovative solutions for enhancing water quality, particularly in challenging contexts like fracking-produced water. Bacterial isolates from livestock wastewater treatment plants (WWTPs), including *Aeromonas* spp. Isolates that were resistant to various antibiotics (penicillin, tetracycline, colistin, and kanamycin), were shown to be sensitive to lytic bacteriophages [152], supporting the potential of phages in biocontrol applications in wastewater treatment and in the prevention of widespread exposure to ARB without producing chemical residues. Indeed, the spread of resistance genes in wastewater treatment plants (WWTPs) has been recognised as a hotspot of resistance gene transfer and supports the hypothesis that alternative treatment approaches are needed [153]. It is equally worth considering “borrowing” from the success of bacteriophage application in areas like food safety. For example, a commercial bacteriophage brand name *Listex*^®^ has found wider use in *Listeria* control [154].

Given the increasing prevalence of antibiotic resistance gene transfer, bacteriophages are gaining attention as a promising method of eliminating pathogens in wastewater treatment plants (WWTPs). While chlorine is widely used in these settings, its effectiveness has notable limitations. Research reported that chlorination failed to eliminate approximately 40% of erythromycin-resistant genes and 80% of tetracycline-resistant genes. Quantitative real-time PCR results further confirmed only modest reductions of 0.42 ± 0.12 log for erythromycin resistance genes and 0.10 ± 0.02 log for tetracycline resistance genes after treatment. Despite these reductions, a significant portion of these antibiotic resistance genes persisted [155], highlighting the need for alternative strategies like phage therapy to more effectively target and reduce antibiotic-resistant bacteria in WWTPs. Consequently, there is an opportunity to exploit phage therapy as a solution to AMR, though this will need to be underpinned by significant ongoing research. Bacteriophage therapy has many benefits, such as its narrow host range (limited to strains of a given bacterial species, typically), which reduces human microbiota effects [156]. Many bacteriophages also have the capability of surviving in unfavourable conditions. One phage isolated from salt ponds in the U.S. was found to tolerate a wider range of environmental conditions than its host *Salicola*, surviving in conditions of pH 3–9 and up to 80 °C [157]

Building on the promising potential of bacteriophages in wastewater treatment and water quality management, recent research has demonstrated their effectiveness against several key waterborne pathogens, including *Klebsiella pneumoniae* and *Pseudomonas aeruginosa*. Phage cocktails specifically targeting Gentamicin-resistant *K. pneumoniae* showed significant lytic activity against both clinical and community-acquired strains, including those resistant to multiple antibiotics, highlighting their utility as effective antimicrobial treatments for nosocomial infections [158]. Similarly, studies on *P. aeruginosa* have identified lytic phages, such as V523, V524, and JG003, with broad host ranges that effectively reduced bacterial counts in water environments. Although bacterial resistance development was noted, phage cocktails provided more consistent reductions and demonstrated the robustness of phages as biocontrol agents across various environmental conditions [83]. Additionally, the presence of antimicrobial-resistant pathogens in healthcare water systems has underscored the necessity for alternative treatment strategies, including phage therapy, to manage biofilm-associated resistance and prevent healthcare-associated infections [36]. However, challenges remain, as illustrated by efforts to mitigate biogenic methanethiol in wastewater, where phage efficacy was reduced under variable environmental conditions, pointing to the need for further optimisation in real-world applications [159]. Together, these findings suggest that while bacteriophages hold significant promise for enhancing water treatment and tackling antimicrobial resistance, there is still a need to carefully consider their application environments and further research to maximise their effectiveness.

A possible application of bacteriophages in water treatment is their use as indicators of pathogenic bacteria and viruses. Faecal indicator bacteria (FIB) are often used to indicate the presence of pathogenic bacteria in water sources or as indicators of water quality, including faecal coliforms and *E. coli*. However, not all pathogenic bacteria strongly correlate with FIB [160] due to their wide range of hosts and their ability to replicate in their environment, which creates inconsistent relationships with pathogens [161]. Indeed, the need for novel viral indicators of faecal pollution in wastewater treatments has been highlighted [123]. This could include the application of bacteriophages. There are currently three main indicator phages: *E. coli* or somatic coliphages, *Bacteroides* phages, and *Enterococcus* phages [160]. Somatic coliphages can infect *E. coli* through their cell walls [161] as they attach to cell wall receptors where replication occurs [162]. F-specific coliphages are often found in domestic water sources and can infect *E. coli* through the adsorption of pili F [162]. This may be underpinned by similar inactivation or “survival” characteristics of coliphages and human enteric viruses due to their similar morphological and structural characteristics compared to bacterial indicator organisms [163].

A study on human health risk of creational water with faecal contamination, the authors proposed that coliphages may be more suitable as indicators of viral pathogens as they closely resemble them [161]. There have also been studies into *Enterococcus* phages that show potential as indicators of faecal pollution and surrogates for enteric viral pathogens in tropical or subtropical climates [125]. This is due to their ability to replicate in a broad temperature range. For example, [164] found that *Enterococcus faecalis* bacteriophages SFQ1 can survive temperatures up to 50 °C but are completely inactivated at 70 °C. The application of *Enterococcus* phages as indicators of faecal contamination in tropical watersheds is gaining attention due to its potential to provide rapid and specific detection of human faecal contamination. Studies have demonstrated that *Enterococcus* phages are highly specific to human faecal contamination and exhibit survival characteristics similar to those of enteric pathogenic viruses, making them suitable candidates for this purpose. They are particularly effective in tropical and subtropical climates, where they have been shown to correlate with environmental factors like rainfall and urbanisation, often presenting higher concentrations in non-urbanized areas. Additionally, the use of *Enterococcus* phages as markers for human faecal pollution in recreational waters and for microbial source tracking has shown promise, highlighting their potential as reliable indices for assessing water quality [165]. Together with the robustness of coliphages, which are already known to survive well in aquatic environments and may serve as suitable indicators of viral pathogens [161,162], the use of phages offers a comprehensive approach to water quality monitoring and the detection of faecal contamination.

It has been found that faecal indicator bacteria (FIB) can survive over 20 days in freshwater in tropical regions. In contrast, Enterophages survive for 3 to 5 days, providing a more accurate snapshot of the microbial communities present at a given time. This observation suggests potential applicability in SSA, where understanding the dynamics of waterborne pathogens is crucial. While the use of bacteriophages as indicators in water treatments has been extensively studied in other regions, there remains a significant gap in research on their application in African settings. A study highlights the research gap and emphasises the need for focused studies in Africa to explore the potential benefits of phage therapy in improving agricultural productivity and managing bacterial infections, particularly those caused by multidrug-resistant strains [166]. By promoting phage-based research, increasing education on bacteriophage biology, and developing tailored phage therapies, SSA could adopt innovative water treatment and public health approaches that address local challenges effectively. This implies that using bacteriophages as indicators of water contamination in Africa is a possible avenue of research to explore. However, bacteriophages are not without their limitations. Although their narrow host range may be of some benefit as it does not harm the human microbiota, as discussed above, it could also be a disadvantage. Lysogenic phages may not lyse all the host bacteria. A study highlighted the risk of phage transmitting toxins and AMG to bacteria [167]. One obstacle to phage therapies faced in many countries is receiving regulatory approval, which could be due to their specificity, which leads them to have a narrow host range. A possible solution for this limitation could be identifying the pathogen before administration or using phage cocktails [168]. Alternatively, phage cocktails may be applied to reduce delivery time. A second limitation of phage therapy is the evolution of bacterial resistance to bacteriophages. This resistance is derived from mutations, CRISPR, and other restriction endonucleases degrading phage DNA [169]. It has been found that sepsis caused by extra-intestinal pathogenic *E. coli* (ExPEC) in mice was reduced by phages, but bacterial numbers remained high due to the development of phage-resistant isolates [170]. A third limitation of phage therapies is the instability of phage in manufacturing and storage. Phages may be inactivated during manufacturing during dehydration and rehydration [171]. Notwithstanding these limitations, phages represent a major resource for replacing or enhancing current antibiotic therapies and sanitation strategies.

## 9. Conclusions

Water quality remains a critical issue in sub-Saharan Africa, where inadequate access to safe drinking water and sanitation significantly impacts public health. Traditional water treatment methods often fail to fully address microbial contamination, exacerbated by environmental factors and anthropogenic activities. Bacteriophages, with their ability to specifically target and lyse bacterial pathogens, offer a promising solution for improving water quality and enhancing public health. This review highlights the potential of bacteriophages as a tool for biocontrol in water treatment and an innovative strategy to combat antimicrobial resistance. By leveraging bacteriophage technology, there is an opportunity to enhance water safety, protect public health, and contribute to achieving global water security goals. In addition, applying high-quality water in food preparation and processing is essential for safe food delivery. By integrating bacteriophage technology into water treatment, communities can benefit from safer food products, further reinforcing the critical connection between water security and food safety. In this way, leveraging bacteriophages in water treatment can contribute significantly to healthier communities, supporting both public health and sustainable development goals in the region.

## Figures and Tables

**Figure 1 viruses-16-01897-f001:**
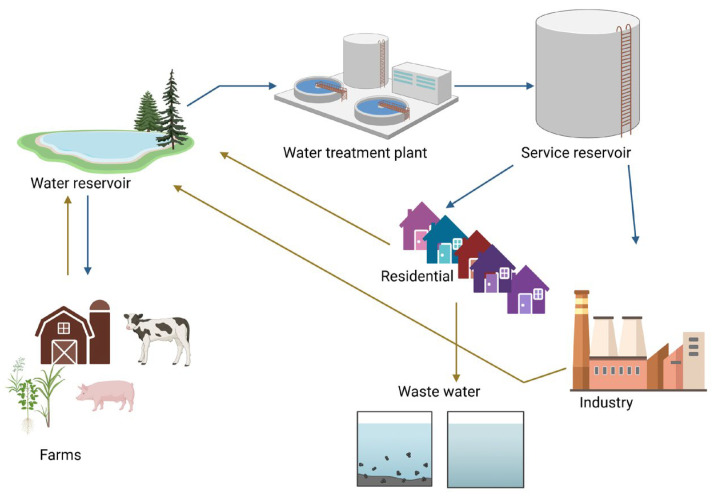
The interaction between water sources, applications, and contamination pathways. Reservoirs supply water to households, farms, and industries, with wastewater from agriculture and industrial activities introducing microbial and chemical contaminants. This figure was created using Biorender.com (accessed on 20 November 2024).

**Figure 2 viruses-16-01897-f002:**
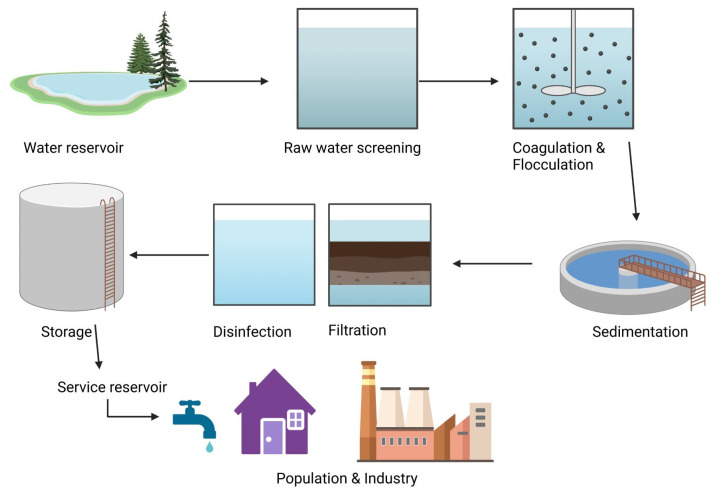
A schematic representation of the water treatment process. Water from reservoirs undergoes screening, coagulation, flocculation, sedimentation, filtration, and disinfection. This figure was created using Biorender.com (accessed on 28 November 2024).

**Figure 3 viruses-16-01897-f003:**
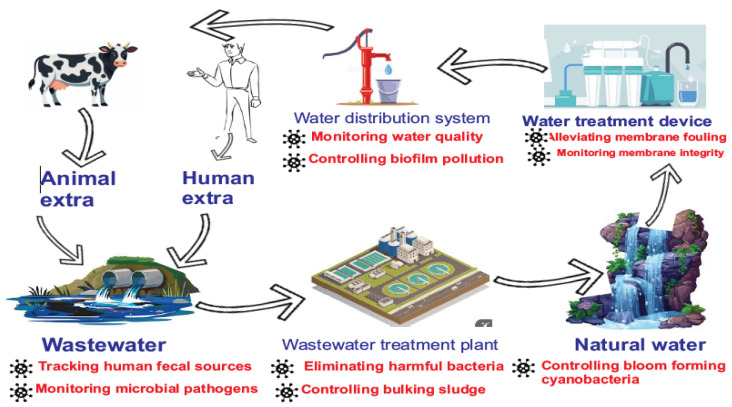
Bacteriophage applications across the water cycle, targeting microbial contaminants in natural water, treatment devices, distribution systems, and wastewater to ensure water quality and safety.

**Table 1 viruses-16-01897-t001:** Bacterial contamination of water sources within sub-Saharan Africa.

Country	Sample and Source	Bacteria	Isolation Technique	Purpose of Study	Reference
Kenya	Lake Victoria	*Acinetobacter* (18.90%), *Arcobacter* (15.12%), *Flavobacteriales* (7.34%), *Dechloromonas* (6.55%), and *Pseudomonas*	Genomic sequencing	Metagenomic analysis of microbial assemblage	[19]
Kenya	Lake Victoria region: water from sewage, ponds, lakes, rivers, and unprotected wells	*Vibrio cholera*	Standard morphological and biochemical tests	Characterisation of *Vibrio cholerae* bacteriophages	[26]
Kenya	Water and fish from the landing beaches of Dunga, Luada, Rombo, and Sirongo in Kisumu, Luanda, and Bondo Municipalities	*Salmonella typhimurium* (49.6%), *E. coli* (46.6%) *V. cholerae* 01 (2.8%)	Standard morphological and biochemical tests	The prevalence and antimicrobial susceptibility patterns of enteric bacteria from Lake Victoria basin	[27]
Kenya	Water samples from shoreline beaches of Lake Victoria Basin, Kenya (Luanda Rombo, Dunga, Osieko, Marengo, Usoma)	*E. coli* (69.6%) *Salmonella* spp. (18.5%) *Shigella* spp. (6.5%) *V. cholerae* (5.4%)	Standard morphological and biochemical tests	An assessment of the physicochemical and bacteriological quality of water within the Lake Victoria Basin in Kenya	[39]
Kenya	Lake Victoria	*Corynebacterium*, *Staphylococcus*, *Turicella*, *Cutibacterium*, *Acinetobacter*, *Micrococcus*, *Faecalibacterium*, *Shewanella*, *Escherichia*, *Klebsiella*, *Enterococcus*, *PrevotellaLegionella*, *Vibrio*, and *Salmonella*	Genomic sequencing	Microbiome analysis of surface and sediment	[18]
Tanzania	Fish intestines, lake water, and fish surface from Lake Victoria basin	*Salmonella salamae* and *S. Waycross*	Genetic sequencing	Investigation of pathogenicity of *Salmonella* spp. isolated from Nile perch	[40]
Uganda	Water samples from Lake Victoria	*Citrobacter freundii* (71%), *Klebsiella pneumonia* 6% (*n* = 31). *Enterococcus* spp. at 77.5% (93/120)	Standard morphological and biochemical tests	Bacteriological analysis of Water and Tilapia fish	[41]
Tanzania	Samples of carp, phytoplankton, and water in Lake Victoria basin	*V. cholerae*	Genetic sequencing	Surveillance and genomics of toxigenic *Vibrio cholerae*	[42]
Kenya	Water samples from pollution hotspots in the Lake Victoria basin, including River Kisat, Chemelil sugar factory effluent, and River Mbogo	*E. coli*, *Enterobacter*, *Salmonella*, *Shigella*, and *Vibrio* spp.	Standard morphological and biochemical tests	Investigation into the association between metal tolerance and multidrug resistance among environmental bacteria from wetlands in the Lake Victoria Basin	[43]
Kenya	Water samples from hyacinth-infested and open-water areas of Lake Victoria, Homa Bay, and Asembo Bay	*E. coli*	Standard morphological and biochemical tests	Assess the effects of water hyacinth (*Eichhornia crassipes*) on the levels of Escherichia coli in Lake Victoria	[44]
Malawi	The shoreline water and beach sand from ten beaches along the southwestern and southeastern arms of Lake Malawi	*E. coli* 17,600 (CFU)/100 mL in sand and 21,200 (CFU)/100 mL in Water	Standard morphological and biochemical tests	*E. coli* distribution in Lake Malawi’s water and sand	[45]
Malawi	Tilapia fish from Lake Malawi	*Pseudomonas* and *Micrococcus* spp., *Shigella*, *Vibrio*, *Acinetobacter*, *Corynebacterium*, *Lactobacillus*, *E. coli*, and *Staphylococcus*	Standard morphological and biochemical tests	Shelf-life analysis of whole fresh Lake Malawi tilapia	[46]
Burundi	Water samples from Lake Tanganyika at Kajaga, Nyamugari, Rumonge, and Mvugo	Faecal coliform bacteria ranged from 0 to 5000 CFU/100 mL, averaging 2000 CFU/100 mL. *E. coli* counts ranged from 0 to 3000 CFU/100 mL, averaging 1350 CFU/100 mL	Standard morphological and biochemical tests	Assessment of coliform bacteria contamination in Lake Tanganyika as indicators of recreational and drinking water quality	[47]
Tanzania	Water samples from Kigoma and Mahale in Lake Tanganyika	*Cyanobacterium:* 30–40% of the surface community. *Proteobacteria* is the most commonly bacterial phyla in freshwater systems. *Actinobacteria*, *Bacteroidetes*, *Verrucomicrobia*, *Thaumarchaeota*, *and Nitrospirae* also identified	Genomic sequencing	Spatial variation of microbial communities	[48]
Tanzania	Water samples from Lake Tanganyika in Kigoma and Mahale regions	*Actinobacteria*, *Alphaproteobacteria*, *Cyanobacteria*, *Verrucomicrobia*, *Euryarchaeota*, *Thaumarchaeota*, and various candidate phyla	Genomic sequencing	Metagenomic of lake microbiome	[49]
Zambia and DRC	Lake Tanganyika water and patient samples	*V.cholerae*	Genomic sequencing	Cholera outbreaks in Lake Tanganyika	[50]
Burkina Faso	Water samples from 39 reservoirs in Burkina Faso	*V. cholera*	Genomic sequencing and culture techniques	Investigate the presence and ecological dynamics of *V. cholerae* in reservoirs in Burkina Faso during the dry season	[51]

**Table 2 viruses-16-01897-t002:** Summary of some alternative water treatment strategies in SSA and actors.

Country	Strategy	Strategy Objective	Actors/Implementers
Kenya, Malawi, and Uganda	Chlorine Dispensers	Treat water at the point of collection	Evidence Action, USAID
Kenya	Community-Led Total Sanitation (CLTS)	Elimination of open defecation	UNICEF Kenya
Uganda	Solar Water Disinfection (SODIS)	Use sunlight to disinfect water	SODIS Foundation
Uganda	Community-Led Advocacy and Action (CLAA)	Educate on water conservation and hygiene	Uganda Ministry of Water and Environment
Ghana	Integrated Water Resources Management (IWRM)	Manage water resources effectively	Ghana Water Resources Commission
Ghana	Ceramic Water Filters	Provide household water purification	Potters for Peace
Malawi	Borehole Rehabilitation	Ensure reliable access to groundwater	Water For People
Rwanda	Piped Water Networks Expansion	Expand access to safe drinking water	Aqua for All
Ethiopia	Water User Associations (WUAs)	Involve communities in water management	Awash Basin Authority
Ethiopia	Water Sector Development Program	Enhance regulatory frameworks and water management	Ethiopian Ministry of Water, Irrigation and Energy
South Africa	Decentralised Wastewater Treatment	Reduce pollution from untreated waste	South Africa Government Report
Tanzania	Solar Water Disinfection (SODIS)	Use sunlight to disinfect water	SODIS Foundation Tanzania
Zambia	Ceramic Water Filters	Provide household water purification	Potters for Peace
Lake Victoria	Real-Time Water Quality Monitoring	Monitor pollution levels and manage resources	Lake Victoria Basin Commission
Nigeria	Mobile Applications for Reporting Water Quality Issues	Enable communities to report water quality	Water Scope
Uganda	EcoSan Toilets	Improve sanitation and reduce source water contamination	EcoSan Uganda
Kenya	Training for Local Water Managers	Build capacity for managing community water points	UNICEF Kenya
Malawi	Solar Water Disinfection (SODIS)	Use sunlight to disinfect water	SODIS Foundation

**Table 3 viruses-16-01897-t003:** Summary of phage studies on water within sub-Saharan Africa. The historic morphological classifications or host range diversity in the absence of genome sequence data was used.

Country	Location	Sample	Study Reference	Phage Morphology	Bacterial Host
Kenya	Lake Elementaita	Water and sediments	[101]	Podovirus	*Ralstonia solanacearum* strain GIM1.74
Kenya	Lake Elementaita	Sediment samples	[96]	Myovirus, Siphovirus, Podovirus	*Vibrio metschnikovii*, *Bacillus* *pseudofirmus*, *Bacillus* *bogoriensis*, *Bacillus horikoshii*, *Bacillus cohnii*, *Bacillus* *psedolcaliphilus*, *Bacillus* *halmapalus*, *Exiguobacterium* *aurantiacum*, *Exiguobacterium* *alkaliphilum*
Kenya	Lake Magadi and Shala	Soil sediments	[98]	Myovirus and Siphovirus	*Bacillus* and *Paracoccus* species
Kenya	Lake Victoria	Water sample	[97]	Myovirus	*Vibrio cholerae*
Ethiopia	Lake Chala	Soil sediments	[98]	Myovirus, Siphovirus	*Bacillus and Paracoccus* species
Tunisia	Kabeli and Tunis	Raw and treated wastewaters	[102]	Somatic coliphages (SOMCPH), F-specific R.N.A. bacteriophages (F-RNA), *B. fragilis* phages (RYC2056), and *Bacteroides thetaiotaomicron* phages	*E. coli*, *Salmonella* *typhimurium*, and *Bacteroides fragilis*
South Africa	Northwest province	Taps, boreholes, and dams	[95]	Myovirus	*V. harveyi*, *V. parahaemolyticus*, *V. cholerae*, *V. mimicus*, *V. vulnificus*
Côte d’Ivoire	Abidjan	Sewage water	[103]	Siphovirus and Podovirus	*Achromobacter xylosoxidans*
Côte d’Ivoire	Cocody area and Treichville area	Sewage samples	[104]	Myovirus, Siphovirus, Podovirus	*P. aeruginosa*
South Africa	Umhlangane River	Water sample	[105]	Myovirus, Siphovirus, Podovirus	Not indicated
Senegal	Dakar peninsula and estuary of Sine-Saloum	Gut and water sample of Tilapia fish	[106]	Myovirus, Siphovirus, Podovirus	Not indicated
Nigeria	Jos, Plateau State	Sewage water	[107]	Myovirus	*P. aeruginosa*
Malawi	Blantyre	Water samples	[108]	Ackermannviridae and Siphovirus	*S. typhimurium*, *S. enteritidis*
Egypt	Zagazig	Sewage	[109]	Siphovirus	*P. aeruginosa*
South Africa	Brandvlei	Hot springs Water sample	[110]	Myovirus, Siphovirus, Podovirus, *Fuselloviridae*	Not indicated
Egypt	Minia	Wastewater sample	[111]	Siphovirus, Podovirus	*E. coli* O104: H4 *E. coli* O157: H7
Tunisia	North and South of Tunisia	Wastewater samples	[102]	Coliphage	Not indicated
Tunisia	MHT	Water samples	[112]	Podovirus	*Klebsiella pneumoniae*
Mauritania		Sewage and water samples	[92]	Myovirus,	*Prochlorococcus* and *Synechococcus* sp.
Kenya		Water samples		Cyanophage	
South Africa	Northwest Province	Portable water samples	[113]	Somatic coliphages and F-RNA coliphages.	*E. coli* strain C (ATCC 700078) and *S. typhimurium* strain WG49
South Africa	Umgeni Water catchment area	Wastewater	[114]	Somatic coliphages and F-RNA coliphages	*E. coli* strain C (ATCC 700078)

**Table 4 viruses-16-01897-t004:** Key phages used for microbial source tracking (MST).

Phage Group	Application in Water Monitoring	Markers for Human Contamination	Marker for Animal Contamination	References
Coliphages	F-RNA coliphages (genogroup I preferred for accuracy)	F-RNA coliphages (genogroups II, III)	F-RNA Coliphage (genogroup IV) MST for cattle	[121,122]
	Somatic coliphages targeting *E. coli* (e.g., WG5)			[123]
Bacteroides Phages	Phages infecting *B. fragilis* (e.g., RYC2056, GB124)	*B. fragilis* phages (e.g., GB124, HSP40)	*B. fragilis* (e.g., PG76) MST for pigs	[124,125]
	*B. thetaiotaomicron* phages (e.g., GA-17, ARABA 84)	Same phages also track human faecal pollution	*B. thetaiotaomicron* phages (e.g., CW18) MST for ruminants, pigs, and poultry	[126,127,128,129,130,131]
	CrAssphage (non-specific, abundant in sewage)	CrAssphage		[132,133]
	Phages of *Bacteroides* HB-73	Same phages used for human source detection	*B. fragilis* PG76 phage MST for pigs	[134,135]
Enterococcus Phages	*E. faecalis* phages (e.g., AIM06, SR14)	*E. faecalis* phages (same strains)		[136,137]
	*E. faecium* phages (e.g., ENT-49, ENT-55)	*E. faecium* phages (e.g., MW47)		[138,139]

## Data Availability

Research data are not shared.

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
