# Peer review of "Exploring the Microbial Ecology of Water in Sub-Saharan Africa and the Potential of Bacteriophages in Water Quality Monitoring and Treatment to Improve Its Safety"

_viruses, 2024, doi:10.3390/v16121897_

Round 1
Reviewer 1 Report
Comments and Suggestions for Authors
The work contains a lot of interesting information, however in my opinion, there is a lack of some information regarding application of bacteriophages in Sub-Saharan Africa water treatment.
e.g., please describe in more detail how the process of water purification with phage is done.
Additionally, I would change the introduction, because the title indicates that the review would be only about bacteria application of bacteriophages in water treatment, and this topic starts at 5 points. Please add information about earlier fragments of the work.
- Figures 1 and 2 - descriptions on drawings are invisible - please correct them, additionally Fig. is too small.
- There is a problem with the page numbering, please check this.
Author Response
Comments and Suggestions for Authors
The work contains a lot of interesting information, however in my opinion, there is a lack of some information regarding application of bacteriophages in Sub-Saharan Africa water treatment.
e.g., please describe in more detail how the process of water purification with phage is done.
Response: Many thanks to the reviewer for their time and consideration of our manuscript. Currently, phages are not applied in water treatment deliberately although they may be naturally present. We propose the potential benefits of such an approach. Section 8 is dedicated to highlighting the potential steps where phages could be added to reduce microbial load and increase the biological safety. This section has also been revised to include a balanced discussion on the potential risks as well as benefits of phage bioremediation and the solutions to the challenges.
Additionally, I would change the introduction, because the title indicates that the review would be only about bacteria application of bacteriophages in water treatment, and this topic starts at 5 points. Please add information about earlier fragments of the work.
Response: Many thanks to the reviewer for highlighting this point. Indeed, the front of the review focuses largely on the ecology of SSA water systems while the latter part deals with the potential application of phages in bioremediation strategies. For this reason, we have amended the title to better reflect the content and it now reads: “Exploring the Microbial Ecology of Water in Sub-Saharan Africa and the Potential of Bacteriophages in Water Quality Monitoring and Treatment to Improve its Safety”.
- Figures 1 and 2 - descriptions on drawings are invisible - please correct them, additionally Fig. is too small.
Response: The figures and font size has now been increased for clarity. Thank you for highlighting this to us.
- There is a problem with the page numbering, please check this.
Response: This has been corrected. Thank you for highlighting this.
Reviewer 2 Report
Comments and Suggestions for Authors
the manuscript titled "Exploring the Potential Application of Bacteriophages in Water Treatment in Sub-Saharan Africa: Towards Improving Food and Water Safety" was well written with sound discovery!
there are some typos error and space gap in the text--please check thoroughly.
1. Figure 1 labeling words font size should be increased
2. is there any side effects of the Bacteriophages treatments on pathogenic bacterial mutations that can causes serious infections towards population-please justify it
Comments on the Quality of English Language
over all good
Author Response
Comments and Suggestions for Authors
the manuscript titled "Exploring the Potential Application of Bacteriophages in Water Treatment in Sub-Saharan Africa: Towards Improving Food and Water Safety" was well written with sound discovery!
Thanks for the supportive comments and suggestions which we have addressed below.
there are some typos error and space gap in the text--please check thoroughly.
Response: The manuscript has been checked thoroughly and carefully revised to ensure that the formatting and grammar are correct.
- Figure 1 labeling words font size should be increased
Response: The font size has now been increased for clarity. Thank you for highlighting this to us.
- is there any side effects of the Bacteriophages treatments on pathogenic bacterial mutations that can causes serious infections towards population-please justify it
Response: The reviewer is correct in highlighting this point. We have mentioned in L560-4 that while phages have significant merit, we should be cognizant of phage-resistance development and as agents of gene transfer as follows: “While studies indicate phages could “solve” the crisis of antibiotic-resistant bacterial prevalence [149], it must not be negated that bacteriophages may contribute to antimicrobial resistance through the transmission of antimicrobial resistance genes (ARGs) by horizontal gene transfer [152]. Therefore, phages selected for therapeutic or bioremediation purposes should be carefully evaluated to minimize such risksand to exclude the application of phages whose genomes harbour antiubiotic-resistance genes or virulence factors, among other traits. Additionally, it should be considered that phage-resistance may develop among sub-populations if repeated applications of the same phages are used. To alleviate this risk, it would be prudent to use large phage cocktails or rotations of different cocktail.